# Point-of-Care Screening for Coeliac Disease in Schoolchildren Reveals Higher Disease Prevalence in Croatia

**DOI:** 10.3390/healthcare11010064

**Published:** 2022-12-26

**Authors:** Mario Mašić, Vera Musil, Tatjana Petričević Vidović, Enida Sičaja, Iva Hojsak, Oleg Jadrešin, Sanja Kolaček, Zrinjka Mišak

**Affiliations:** 1Children’s Hospital Zagreb, Referral Centre for Paediatric Gastroenterology, Hepatology and Nutrition, 10000 Zagreb, Croatia; 2University of Zagreb, School of Medicine, 10000 Zagreb, Croatia; 3Andrija Štampar Teaching Institute of Public Health, Department of School and Adolescent Medicine, 10000 Zagreb, Croatia; 4Josip Juraj Strossmayer University of Osijek, School of Medicine, 31000 Osijek, Croatia

**Keywords:** coeliac disease, epidemiology, child, point-of-care testing

## Abstract

Coeliac disease (CD) is an immune-mediated inflammatory disease triggered by dietary gluten and related proteins in genetically predisposed individuals. Point-of-care (POC) methods are non-invasive and easily performed tests, which could help to reduce the diagnostic delay of CD. The aim of our study was to determine the prevalence of CD using rapid POC test in first-grade schoolchildren in Zagreb, Croatia. A rapid qualitative immunoassay POC test designed for detection of immunoglobulin (Ig) A and IgG deamidated gliadin antibodies (DGP), as well as total IgA (to identify IgA deficient patients) in whole blood, was used to test healthy children on gluten containing diet. Out of 1404 tested children (51% female), 85 (6.05%) had a positive rapid POC test result and were referred to paediatric gastroenterologist. Finally, 7 children were diagnosed with CD (0.5%). There was no significant difference in children with CD and children with positive POC but negative serology in sex, BMI, or symptoms. However, children diagnosed with CD complained of abdominal pain significantly more often. The prevalence of CD in first-grade schoolchildren was 1:200 (0.5%), higher than in previous studies performed in Croatia. The results imply the possible benefit of IgA and IgG DGP-based POC tests in population screening.

## 1. Introduction

Coeliac disease (CD) is a chronic small intestinal immune-mediated enteropathy triggered by dietary gluten and related proteins found in wheat, rye, and barley, occurring in genetically predisposed individuals, those with HLA (Human Leukocyte Antigen) DQ2 and/or DQ8 haplotypes. In patients with CD, ingestion of dietary gluten results in autoimmune intestinal mucosal damage characterised by villous atrophy, crypt hyperplasia, and an increased number of intraepithelial lymphocytes [1,2,3,4]. Patients with CD may present with a wide range of symptoms or may be asymptomatic. Gastrointestinal symptoms and signs include chronic or intermittent diarrhoea, weight loss, failure to thrive, nausea or vomiting, chronic abdominal pain, cramping or distension, chronic constipation, and abnormal liver biochemistry. There is also a broad spectrum of possible extraintestinal symptoms, such as dermatitis herpetiformis, stunted growth, delayed puberty, amenorrhoea, recurrent aphthous stomatitis, iron-deficiency anaemia, fracture with inadequate traumas, irritability, arthritis, and chronic fatigue [2,3,4,5]. There is an evident shift from gastrointestinal to extraintestinal symptoms in children, and recent data suggest that the frequency of those symptoms is similar in adults and children [6,7]. Patients with autoimmune diseases and chromosomopathies, such as diabetes mellitus type I, autoimmune liver disease, autoimmune thyroid disease, Down syndrome, Turner syndrome, William’s syndrome, and patients with IgA deficiency have a higher risk of developing CD and should be screened periodically. First-degree relatives of patients with CD are also a high-risk group in whom active screening should always be conducted [4]. The diagnostic procedure involves the assessment of coeliac disease-specific antibodies, such as antibodies to tissue transglutaminase (tTG), endomysial antibodies (EMA), and immunoglobulin G (IgG)-deamidated gliadin peptides (DGP) or other IgG type antibodies in patients with IgA insufficiency, and sometimes an endoscopic procedure with small bowel biopsies. In children, according to the European Society for Paediatric Gastroenterology, Hepatology and Nutrition (ESPGHAN) guidelines, under certain conditions, the diagnosis can be established with a strongly positive tTG antibody titre, if confirmed with positive EMA antibodies in a second blood sample, without the need for small intestinal biopsy [8]. The therapy for confirmed CD is a strict lifelong gluten-free diet which results in a full clinical and histological remission occurs, with reduced morbidity and mortality [9,10]. 

Standard serology testing is used to establish a diagnosis, but it is only available at specialised centres, especially EMA antibodies. New and improved point-of-care (POC) methods are non-invasive and could help reduce diagnostic delay in asymptomatic individuals, especially in areas with limited access to laboratory-based testing [11,12,13]. Although there are many promising results on POC testing, they are still not included in diagnostic algorithms for CD, mostly due to conflicting results on specificity and sensitivity of various POC tests, especially in asymptomatic children [12,14]. 

The global prevalence of CD is 1% and differs among countries. In the United States, the overall prevalence is 1%, significantly higher in Caucasians. In Europe, the highest prevalence of CD is found in Sweden (3%) and Finland (2.4%), while it is lower in Italy (0.7%), Western Ireland (0.6%), and Germany (0.3%) [3,15,16,17,18]. There are limited data on CD prevalence in Croatia, determined by research conducted 20 years ago in one region of the country. The cumulative incidence was found to be 1.9:1000 live births and a prevalence of 0.21% [19,20]. A more recent study performed in Croatia tried to determine the prevalence of CD in first-grade school children using the IgA anti-tTG POC test, and not a single child with CD was detected [21]. 

The aims of our study was to determine the prevalence of CD using the rapid IgA and IgG DGP POC test in first-grade school children and to asses the characteristics of children found by POCT screening, such as sex, body mass index (BMI), and the presence of symptoms. 

## 2. Materials and Methods

### 2.1. Participants

This cross-sectional study was conducted on a population of first-grade elementary school-children for the school year 2018/2019 in the city of Zagreb, born in 2011 and 2012 (ages six and seven years old). The screening was offered to a sample of 2000 children, who comprised 15.8% of the first-grade elementary-school population in city of Zagreb. Overall, 1404 children (11% of elementary-school population in Zagreb) and their parents or legal guardians agreed to screening procedure. The screening test was performed in the Department of School and Adolescent Medicine, Andrija Štampar Teaching Institute of Public Health, during a mandatory visit before enrolment in first grade of elementary school. The exclusion criteria for participation in this study were as follows: previously diagnosed CD, the exclusion of gluten from diet for other reasons, and age older than 7 years. None of the children had previously diagnosed CD. All children included in screening were otherwise healthy and consumed gluten without restrictions. 

### 2.2. Methods

POC-test SimtomaX, a lateral-flow immunochromatographic test designed for detection of immunoglobulin (Ig) A and IgG deamidated gliadin antibodies (DGP) as well as total IgA (to identify IgA deficient patients), was used for screening. According to the manufacturer, the test has been validated with blood samples obtained through finger pricking and venepuncture. In our patients, the first method was used. The test requires a small amount of whole capillary blood from a finger prick, and the blood is placed on a test. The tests were performed according to the manufacturer’s instructions by previously instructed staff in the department of school and adolescent medicine, school doctors, and school nurses. The tests were read by school doctors after 15 min of exposure time. A synthetic DGP is conjugated at test line A for detection of IgA and IgG anti-DGP. If there are DGP antibodies in the patient’s blood, a detectable complex is formed. At test line B, mouse antihuman IgA allows total IgA detection, identifying possible false-negative results caused by IgA deficiency. The test is considered invalid if the control line does not appear, e.g., if the buffer does not diffuse properly onto the strip. A CD-positive test result is indicated by detection of both the control and A lines. The absence of the B line indicates IgA deficiency. Various assessments have been conducted for this commercial kit, and it has been proved to have high sensitivity (95–100%) and specificity (93.1–95.7%) [22,23]. Children with a positive POC test result were referred to the paediatric gastroenterology department at Children’s Hospital Zagreb for clinical examination and further diagnostics (total IgA and coeliac serology testing-IgA tTG in IgA sufficient children and IgG DGP in those with IgA deficiency) (Figure 1). At clinical examination conducted by a paediatric gastroenterologist, we obtained data about the presence of CD symptoms and family history for CD, associated diseases, body weight and body height. BMI was calculated as a ratio of body weight in kilograms (kg), and body height in meter (m)^2^. The diagnosis of CD was established according to diagnostic guidelines of the European Society for Paediatric Gastroenterology, Hepatology, and Nutrition (ESPGHAN) [4,8].

### 2.3. Ethical Considerations

The principle of the test was explained to parents and caregivers by school doctor, and their consent was obtained. Ethical committees of Children’s Hospital Zagreb and Andrija Štampar Teaching Institute of Public Health approved the study.

### 2.4. Statistical Analysis

The differences between categorical variables were assessed by chi-square test, and the differences for non-categorical variables were assessed by ANOVA with *p* < 0.05 indicating that there is a statistically significant difference between the examined groups. Statistical analysis was performed using SPSS software (SPSS Inc., Chicago, IL, USA). 

## 3. Results

There were 1404 children (51% girls, average age 7.23 years) included in the study and tested with POC. The results of the POC tests are as follows: for 1307 (93.1%) children, POCT was negative and showed normal total IgA and negative DGP. Out of the remaining children, 85 children (6.05%) had positive POCT (with normal total IgA and positive DGP), and 12 children (0.85%) were IgA-deficient but DGP-negative. None of children tested with POC were negative for total IgA but positive for DGP (Figure 1).

Children with a positive rapid POC test result (n = 85; 6.05%) were referred to paediatric gastroenterology. At appointment, a paediatric gastroenterologist obtained data about presence of CD symptoms and family history for CD, associated diseases, body weight, and body height. BMI was calculated as a ratio of body weight in kilograms (kg) and body height in meter (m)^2^. They were all clinically examined and advised to perform CD serology testing (total IgA and anti-tTG). Caregivers of four children refused to do further diagnostic testing and two were excluded because they were older than first-graders (those were symptomatic older brothers of two included patients). Out of 79 POC positive children referred for further testing, 8 had positive IgA antibodies against tTG. Concerning the children with IgA insufficiency (regarded as levels of IgA less than 0.2 g/L), we used IgG-DGP antibodies, and all IgA insufficient children were negative for CD. Finally, seven children were diagnosed with CD (0.5%) and one child who had normal intestinal mucosa on intestinal biopsy samples was diagnosed with a potential CD. Six children (86%) with CD had anti-tTG ≥ 10 times the upper limit of normal (ULN), and one had 1.5 times the ULN. Six children underwent upper endoscopy (one had Marsh 2, and two 3a, one 3b, and two 3c), and one child was diagnosed with no-biopsy approach according to ESPGHAN guidelines (Table 1) [8].

Out of 71 POC-positive children but negative for CD, 32 children (45%) had the following symptoms: abdominal pain (20), constipation (10), prolonged diarrhoea (2), frequent vomiting (2), recurrent aphthous ulcers (2), and one failure to thrive. In children later diagnosed with CD, four children had symptoms, all four having recurrent abdominal pain, plus one who also had constipation and one who had frequent aphthous ulcerations. We compared CD children with those who were positive for POC but had negative anti-tTG, and we did not find any significant differences in age nor BMI. Children diagnosed with CD were not symptomatic more often than those who were positive for POC but had negative serology (Table 2). However, when compared regarding only abdominal pain, children diagnosed with CD complained of abdominal pain significantly more often (*p* = 0.002). 

## 4. Discussion

CD is a global, common disease with a rising prevalence in the last 50 years and a broad spectrum of disease manifestations, affecting approximately 1% of the population [24,25,26]. However, the increasing prevalence is most probably due to case-finding-screening strategy, better serological tests, and raised awareness of physicians [27]. CD is only relatively uncommon in Southeast Asia and sub-Saharan Africa, while there are reports that it is under-recognised in China [24,28,29]. The prevalence of CD is also low in some European countries, such as Estonia, even despite a dramatic increase in incidence during the last 35 years [30]. The two major determinants of CD prevalence are HLA haplotype and wheat consumption, but other risk factors appear to be in relation to the development of CD, such as environmental factors, geographic region, ethnicity, and origin [31,32]. The data on CD prevalence and risk factors in Croatia are limited. One study performed in a healthy high-school-student population showed the prevalence of 0.21%, and another, which used the IgA anti-tTG POC test, did not find any cases in more than 1400 of 7-year-old children [20,21]. In our study, the prevalence of CD in a population of first-grade children was 1:200 (0.5%) individuals, which is considerably higher than that previously reported. The prevalence in our cohort could have been higher, as the study did not include blood serology tests on POC-negative individuals, and we cannot exclude the possibility of missing CD patients due to possible false negative POC tests. 

This study used the SimtomaX Blood Drop test to detect immunoglobulin IgA and IgG antibodies against DGP. According to studies, the test showed good high sensitivity (95–100%) and specificity (93.1–95.7%) in ruling out symptomatic CD individuals [22,23]. In this study, we had a significant number of false-positive tests, and we can only speculate that the patients who had faint lines in their test were also referred to paediatric gastroenterology, as a similar experience was shown in other studies [33]. As other researchers suggest, with a large number of false-positive tests, as was our experience, the role of the test cannot replace conventional serology, but it can be used as a triage test to decide if conventional serology is necessary [34]. However, the opinions of researchers on determining the prevalence of CD with rapid POC tests are not unilaterally positive. In study by Dekanić et al., 1487 children were screened, and not a single child with CD was detected; thus, the authors concluded that the POC test was not useful. The researchers used a different lateral flow test determining the presence of endogenous tTG IgA and total IgA, which, according to existing research, also showed a good sensitivity and specificity in different age groups [21,35]. Other studies also showed lower sensitivity and specificity of rapid POC tests than expected, resulting in a lower CD prevalence in examined populations [36]. According to researchers, proper training in interpreting the results could improve the strength of the test [11,21]. All of the presented data suggest that POC tests could be used to identify patients with undiagnosed CD, but further studies and validation are required in finding the optimal test, especially in the paediatric population. 

According to our study’s findings, 43% of children detected by this screening and diagnosed with CD were asymptomatic and otherwise would not be diagnosed with CD. Our results are in concordance with global trends, as the data show that 90% of the patients remain under-diagnosed or undiagnosed in clinical practice due to being asymptomatic or oligosymptomatic [2,11,12]. Out of symptomatic patients (57%), all children had mild, recurrent abdominal pain, and one patient had constipation and another recurrent aphthous ulcers. As the symptoms were mild and not of a concern to parents, they were not investigated by primary physician. Although clinically silent, our patients had significant intestinal mucosal lesions. Our study results are also in agreement with global trends, highlighting the importance of diagnosing clinically silent patients. Recent studies and reviews suggest that patients diagnosed later in childhood have mild or non-existing gastrointestinal symptoms. In addition, when the disease manifests, the clinical symptoms are often extraintestinal and monosymptomatic [27]. Our results are in agreement with those of other studies concerning the clinical presentation of coeliac disease in this region of Europe. As per study by Riznik et al., from data on 653 children and adolescents in Central Europe (including Croatia), the most common symptom was abdominal pain in monosymptomatic and polysymptomatic children, especially in the older group of children. [37]. We recognise that our small cohort of diagnosed CD patients in this study makes it difficult to recognise true differences with POCT-negative children, but nonetheless, the results are still in agreement with those other studies describing clinical characteristics of CD patients in this population [37,38]. 

Complications of CD are common, and CD is known to have long-term complications and an increased risk of malignancy and mortality. There is evidence that untreated disease is associated with delayed puberty, epilepsy, osteoporosis, behavioural disturbances, reduced educational performance, and malignancies (small intestinal adenocarcinoma and T-cell lymphoma) [11,39,40]. Although there are only few studies regarding the long-term health-economic aspect of CD, complications and morbidity present an important socio-economic factor, and timely CD diagnosis is associated with a significant overall healthcare cost reduction [40,41]. 

Screening for CD fulfils most of the criteria for mass screening (prevalence, known morbidity and disease complications, and available treatment), but due to the poor understanding of natural course of the disease and the outcome of asymptomatic patients, there are still no recommendations for mass screening, except in high-risk groups [42]. There is also debate as to whether population screening for CD is beneficial regarding the compliance to diet and quality of life, but studies report that a gluten-free diet improved quality of life in both symptomatic and asymptomatic screened populations and that the adherence to a gluten-free diet was good after the diagnosis was established by screening [43,44]. 

Active screening is important in high-risk groups, such as first-degree relatives and in individuals with autoimmune diseases and developmental diseases, such as diabetes mellitus type 1, autoimmune liver disease, autoimmune thyroid disease, Down syndrome, Turner syndrome, William’s syndrome, and individuals with IgA deficiency, who have a higher risk of developing CD. Those patients should be actively screened for CD periodically [2,3,4,45]. Other studies have also shown the importance of active screening of the risk groups and asymptomatic screening as almost 20% of the newly found CD patients can be found with the aforementioned methods [37]. 

There is a clear need for a rapid, cheap, and discriminative POC test that could facilitate the triage of suspected CD patients. The aforementioned is mostly crucial in areas with no availability to coeliac-specific serology testing, especially EMA antibodies, which are determined in only specialised laboratories. The proposed test should have optimal specificity and sensitivity, or it should have minimal false negative results while still being cost effective. Those tests could be most useful for primary care physicians and could help relieve the referrals to secondary and tertiary health care institutions. Furthermore, there is a clear need for education about coeliac disease among patients and healthcare professionals, which could altogether reduce the delay in diagnosis [46].

This study has potential limitations. The data on the incidence and prevalence of CD in Croatia are limited and rely on the results of one study in a limited region of country. Our study aimed to determine the prevalence (using a rapid POC test) and clinical characteristics of CD found by screening among first-grade schoolchildren in Zagreb, Croatia. As the prevalence of CD could differ among other regions in Croatia, we cannot estimate the overall prevalence in Croatia, and there is a need for further research. Croatia has interesting geographical features, with a Mediterranean coastline, large number of islands, and classical continental region; as such, it would be interesting to study possible differences in CD prevalence in those regions. However, approximately one quarter of Croatia’s population is centralised around Zagreb County. Furthermore, this POC test is reported as highly sensitive, but as not all children screened with POC had performed the serology test, we could not determine whether there were false negative POC results and, therefore, missed CD patients. We also screened seven-year-old children, and there is a debate as to what age is appropriate for the timing of screening. Finally, as the test we conducted is qualitative, an unintentional error could have occurred in reading the test results, as a faint positive line could have been missed. 

## 5. Conclusions

In our study, the prevalence of CD in first-grade schoolchildren was overall 1:200 (0.5%). Children diagnosed with CD did not have significantly more symptoms, thus representing the clinically silent spectrum of coeliac disease patients. In this study, we could not determine whether there were false-negative tests, thus resulting in missed CD patients. However, this is a higher prevalence of CD than reported earlier for Croatia, showing the benefit of IgA and IgG DGP-based POC tests in population screening.

## Figures and Tables

**Figure 1 healthcare-11-00064-f001:**
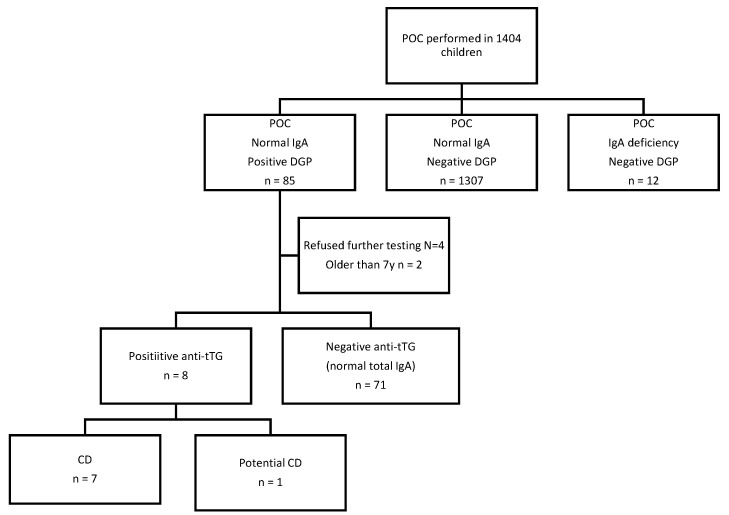
Flow chart of the screening.

**Table 1 healthcare-11-00064-t001:** Characteristics of children diagnosed with coeliac disease (CD).

	Age	Sex	BMI z-Score	Symptoms	IgA	tTG	EMA	IgG DGP	Marsh
Case 1	6y 7mo	F	0.25	no	1.56	>128		14.18	2
Case 2	7y 5mo	M	−1.53	constipation, abdominal pain	0.82	>128		23	3c
Case 3	7y 4mo	F	1.93	abdominal pain, recurrent aphthous ulcers	0.64	>128		48	3a
Case 4	7y 7mo	F	0.67	no	1.62	>128	3+	120	no-biopsy approach
Case 5	7y 3mo	F	0.78	no	0.78	>128		164	3c
Case 6	7y	M	−1.44	abdominal pain	1.71	15		7.8	3a
Case 7	6y 10mo	M	0.35	abdominal pain	1.67	124		29	3b

**Table 2 healthcare-11-00064-t002:** Comparison of children diagnosed with coeliac disease (CD) vs. children with positive point-of-care test but excluded CD (negative serology).

	Confirmed CD (n = 7)	Negative Serology–CD Excluded (n = 71)	*p*
Gender (female vs. male)	4:3	39:32	*p* = 0.78
Age (years) (median, range)	7.08 (6.58–7.42)	7.25 (6.58–8.67)	*p* = 0.43
BMI (median, range)	16 (13.8–21.3)	15.36 (13.8–21.6)	*p* = 0.13
Proportion of symptomatic children	4/7 (0.57)	32/71 (0.45)	*p* = 0.51
Abdominal pain	4/7 (0.57)	20/71 (0.28)	*p* = 0.02
Positive family history	0	1	
Associated disease	0	0	

## Data Availability

The data are available in group form by request to the corresponding author.

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
