# Peer review of "Point-of-Care Screening for Coeliac Disease in Schoolchildren Reveals Higher Disease Prevalence in Croatia"

_healthcare, 2022, doi:10.3390/healthcare11010064_

Round 1

Reviewer 1 Report

Dear Authors,

Thank you very much for submitting your work, and giving me the opportunity to review the manuscript "Point-of-care screening for coeliac disease reveals higher dis ease prevalence in Croatia".

The information provided in the manuscript is interesting and provides information about the prevalence of coeliac disease detected by point-of-care testing for DGP antibodies in school-aged children.

Results of the study show that the prevalence of CD is higher than expected in schoolchildren in Zagreb, Croatia as detected by DGP antibodies. Such information can influence early CD detection strategies.

There are some issues that need to be explained in more detail or need clarification and need to be addressed by the authors, which could possibly improve the quality of the work.

GENERAL COMMENTS

It is interesting that a rather high proportion of tested children were found to be positive with rapid test, who were later found seronegative.

Since only the positive POCT children were tested further, the true prevalence could differ from one described by authors, since POCT could have been false negative in some patients.

SPECIFIC COMMENTS

Title

Possibly include the description of the population in the title: Schoolchildren.

Abstract

Should reflect any changes in the main text.

Keywords

appropriate

Introduction/Background

 Patients with autoimmune diseases and developmental diseases

Consider rephrasing: chromosomopathies

Consider listing first degree relatives as a specific group.

The current diagnosis is based on demonstrating the enteropathy in small 48 intestinal biopsies and the presence coeliac disease

This is not entirely reflecting the latest ESPGHAN guidelines (2012, 2020), where enteropathy is assumed with very hight titres of disease specific antibodies.

The 2020 ESPGHAN guidelines are also not suggesting the use of IgA DGP Abs.

Standard serology testing is used to establish a diagnosis, but it is expensive and only available at specialized centres.

Consider using only the availability issue. Price of serology is not that high, however in some regions the insurance does not cover for serology tests, and in some regions these tests (especially EMA) are not available.

The global prevalence of CD is 1% and differs among countries. In the United States, the…

Consider using a new paragraph.

the highest prevalence of CD is in 65 Sweden (3%) and Finland…

is found in Sweden…

Consider citing Roberts: Roberts SE, et al. Systematic review and meta-analysis: the incidence and prevalence of paediatric coeliac disease across Europe. Aliment Pharmacol Ther. 2021

… anti-tTG POC test and not even one child with CD was detected

Consider using …not a single child was detected

Aim

The aim of our study was to determine the prevalence of CD using rapid IgA and IgG DGP POC test in first-grade school children and to correlate it with sex, body mass index (BMI) and presence of symptoms.

Not entirely clear regarding the correlation part of the aim. Authors were trying to assess any differences re studied parameters between POCT positive, POCT negative and seropositive children and were not correlating the prevalence with these parameters. Consider paraphrasing the aims.

Methods

Were previously diagnosed CD children fully excluded from the prevalence study. If they were included the prevalence figures might be different (higher).

The diagnosis of 110 CD was established according to diagnostic guidelines of the European Society for Paediatric Gastroenterology, Hepatology, and Nutrition (ESPGHAN) [8].

Please provide the data whether all these children were diagnosed using intestinal biopsy or not.

Results

Can you include the number of previously diagnosed CD patients?

for 1307 children test showed normal total IgA…

Please include % of total for each subgroup.

85 children had normal total IgA and positive anti tTG and 12 children were IgA deficient but DGP negative.

Not clear. Were 85 found to be TGA positive?

What about 12 IgA deficient and DGP negative. Were they POCT DGP negative or serology negative for IgG DGP? Make this clearer.

were referred to paediatric gastroenterology….

Please describe as in the methods.

none was IgA deficient but 8 had positive IgA antibodies…

No need to describe total IgA status of this group as it is assumed to be normal by POCT already. Just keep the TGA IgA status.

Finally, 7 children were diagnosed with CD

Include the information about the biopsy outcome in the text.

Consider not using negative biopsy. Instead use normal intestinal mucosa.

Have you been able to check the symptoms in POCT negative participants? To compare them with POCT positive seropositive and POCT positive seronegative patients.

Can you elaborate in more detail on IgA deficient children.

Discussion

Discussion is perhaps too long. Consider a more focused approach on main findings of the study.

Cite Roberts 2021

The prevalence of CD was also regarded as low in some European countries, such as Estonia, although recent research suggests a dramatic increase in incidence of CD in Estonia during the last 35 years, albeit still low in comparison to other European countries [29].

Maybe condense slightly.

In our study, the prevalence of CD in a population of first-174 grade children was 1:200 (0.5%) individuals…

This is the prevalence as detected with POC testing. However true prevalence might be higher if some children had been already diagnosed with CD previously.

Also, false negative DGP could play a role, which is however unlikely, due to high sensitivity of the test. Authors are nicely elaborating on this limitation of the study.

… and not even one child with CD was detected

And not a single child was…

Those patients were not in treatment by a primary physician, as their symptoms were mild…

Slightly unclear

Recent studies and reviews suggest that patients diagnosed later in childhood have mild or non-existing gastrointestinal symptoms. In addition, when the disease manifests, the clinical symptoms are often extraintestinal and monosymptomatic [26].

Consider citing a recent paper re clinical presentation of CD presenting the data from the same region as the current study

Riznik P, et al. JPGN 2021

Active screening is important in high-risk groups, such as first-degree relatives and in individuals with autoimmune diseases and developmental diseases such as diabetes mellitus type 1, autoimmune liver disease, autoimmune thyroid disease, Down syndrome, Turner syndrome, William's syndrome, and individuals with IgA deficiency, which have a higher risk of developing CD.

Same as above.

Limitations are disclosed nicely by the authors.

Conclusions

Comment re the prevalence as stated above. True prevalence should include previously diagnosed CD children.

Literature

Consider checking for the available literature focusing on the same issue published within the last two/three years.

The ESPGHAN 2020 paper is the only one citation used after 2019.

Tables

Appropriate.

Consider shortening the text in the Results that is very nicely presented in the Table one to avoid the duplication.

Figures

Appropriate flow chart.

Consider adding data on previously diagnosed CD patients within the tested cohort.

Author Response

Mario Mašić, MD

Referral Centre for Pediatric Gastroenterology and Nutrition

Children’s Hospital Zagreb

Klaićeva 16, Zagreb, HR-10000, Croatia

Mail: mmasic2@gmail.com

Healthcare

December 16th, 2022

Dear Reviewer,

We are sending our revised manuscript entitled “Point-of-care screening for coeliac disease in schoolchildren reveals higher disease prevalence in Croatia” for possible publication in the journal Healthcare. We would like to thank you for your useful comments, which are all addressed in detail below. We hope that you will find the revised manuscript suitable for publication.

Please find the addressed comment in the attachment below. 

Kind regards,

Mario Mašić

Reviewer 2 Report

This study aimed to define the prevalence of coeliac disease among children in first grade in Croatia. The utility of this research is to assess whether a POC method such as DGP through finger pricking. The investigators found that the prevalence of 0.5% found in this study sample was higher than previous reports in Croatia. 

I believe this study offers some value in highlighting how widespread screening tools could be used for case identification of coeliac disease. In particular, the value would be to help identify asymptomatic individuals or those who do not get tested in other healthcare settings based on non-specific symptoms. Therefore, I think there should be more emphasis on the real-world implications of how a rapid POC test could be implemented into population settings, and further discussion as to the benefits/risks associated with widespread screening. The discussion covers several interesting points but it is not organized in a clear way for the reader to follow the main take-aways from the study. 

There are a few other points worth addressing:

1. Individuals with known coeliac disease were excluded. Are the authors able to report an overall prevalence of coeliac disease with both previously diagnosed CD and newly identified? Similar to this recent study from Norway: https://pubmed.ncbi.nlm.nih.gov/35879335/

This would then provide an overall estimate of CD prevalence in the region and also indicate how many cases are at-risk of remaining undiagnosed. 

2. Were there children who declined to be screened altogether? If so, it would be important to include how many declined to participate to understand if it would have any possible effect on the estimates. 

3. Table 2 should included the exact p-values for the non-significant tests. 

4. The authors should discuss how a very small sample (N = 7) makes it difficult to adequately test differences between groups. Even though most findings were not significant, greater sample size would have increased statistical power to detect any true differences. 

5. The authors note that the region here may not be generalizable to the rest of Croatia. Further discussion as to how the population studied is similar/different from the entire nation. 

Author Response

(The authors gave the same response as above.)
